# Nature-Based Solutions for the Sustainable Management of Urban Soils and Quality of Life Improvements

Slaveya Petrova [1,2,*], Iliana Velcheva [1], Bogdan Nikolov [1], Nikola Angelov [1], Gergana Hristozova [3], Penka Zaprjanova [4], Ekaterina Valcheva [4], Irena Golubinova [5], Plamen Marinov-Serafimov [5], Petar Petrov [6], Veneta Stefanova [6], Evelina Varbanova [7], Deyana Georgieva [7], Violeta Stefanova [7], Mariyana Marhova [8], Marinela Tsankova [8] and Ivan Iliev [8]

1 Department of Ecology and Ecosystem Conservation, Faculty of Biology, University of Plovdiv "Paisii Hilendarski", 24 Tzar Asen Str., 4000 Plovdiv, Bulgaria; anivel@uni-plovdiv.bg (I.V.); nikolov81bg@uni-plovdiv.bg (B.N.); n.angelovv@uni-plovdiv.bg (N.A.)
2 Department of Microbiology and Ecological Biotechnologies, Faculty of Plant Protection and Agroecology, Agricultural University, 12 Mendeleev Blvd., 4000 Plovdiv, Bulgaria
3 Department of Educational Technologies, Faculty of Physics and Technology, University of Plovdiv "Paisii Hilendarski", 24 Tzar Asen Str., 4000 Plovdiv, Bulgaria; ghristozova@uni-plovdiv.bg
4 Department of Agroecology and Ecosystem Protection, Faculty of Plant Protection and Agroecology, Agricultural University, 12 Mendeleev Blvd., 4000 Plovdiv, Bulgaria; p_zapryanova@au-plovdiv.bg (P.Z.); e_valcheva@au-plovdiv.bg (E.V.)
5 Agricultural Academy, Institute of Forage Crops, 89 General Vladimir Vazov Str., 5800 Pleven, Bulgaria; irena.golubinova@ifc-pleven.org (I.G.); plamen.serafimov@ifc-pleven.org (P.M.-S.)
6 Department of Ecology, Environmental Protection and Restoration, Faculty of Ecology and Landscape Architecture, University of Forestry, 10 Kliment Ohridski Blvd., 1756 Sofia, Bulgaria; ppetrov@ltu.bg (P.P.); vstefanova@ltu.bg (V.S.)
7 Department of Analytical and Computer Chemistry, Faculty of Chemistry, University of Plovdiv "Paisii Hilendarski", 24 Tzar Asen Str., 4000 Plovdiv, Bulgaria; varbanova@uni-plovdiv.bg (E.V.); georgieva@uni-plovdiv.bg (D.G.); stefanova@uni-plovdiv.bg (V.S.)
8 Department of Biochemistry and Microbiology, Faculty of Biology, University of Plovdiv "Paisii Hilendarski", 24 Tzar Asen Str., 4000 Plovdiv, Bulgaria; mmarhova@uni-plovdiv.bg (M.M.); marinela_89@uni-plovdiv.bg (M.T.); iziliev@uni-plovdiv.bg (I.I.)
* Correspondence: slaveya_petrova@uni-plovdiv.bg; Tel.: +359-890-933-955

**Abstract:** The rehabilitation and restoration of land-based ecosystems is a key strategy for recovering the services (goods and resources) ecosystems offer to humankind. The use of nature-based solutions (NBSs) to restore degraded soil functions and improve soil quality can be a sustainable and successful strategy to enhance their ecosystem services by working together with the forces of nature and using well-designed measures that require less maintenance, are more cost-effective, and if constructed in the right way may even be more effective over long periods because nature's forces can increase the structural efficiency. In this study, we aimed to (i) evaluate the bioremediation capacity of some grasses and their suitability for lawn planting in settlements (in residential and non-residential areas, along roads, etc.) and (ii) propose technological solutions for their practical application in an urban environment. Emphasis was placed on the potential of some perennial grasses and their application for the bioremediation of polluted urban soils, including perennial ryegrass (*Lolium perenne* L.), crested wheatgrass (*Agropyron cristatum* L.), tall fescue (*Festuca arundinacea* Schreb), and bird's foot trefoil (*Lotus corniculatus* L.). A case study from the city of Plovdiv (Bulgaria) is presented, together with an effective technological solution for the establishment of urban lawns and the roadside green buffer patches.

**Keywords:** phytoremediation; soil pollution; perennial grass; potentially toxic elements; ecosystem services; buffer green patches

## 1. Introduction

Urbanization and industrialization are the main drivers of rapid land-use changes around the world, combined with significant disturbances in soil quantity and quality [1]. Urban soils are formed under the dynamical impacts of both natural soil formation factors and anthropogenic factors [2]. Soil degradation is often observed in urban areas and is expressed as a complete or partial loss of individual soil functions due to increased levels of potentially toxic elements (PTEs), acidification, compaction, and biodiversity loss.

PTEs in soil originate from various natural and anthropogenic sources such as soil-forming minerals and rocks, waste disposal, sewage sludge application, agrochemicals use, mining and smelting, and industrial effluents. [3]. When regarding urban soil pollution with PTEs, their anthropogenic inputs are mainly attributed to the wet and dry atmospheric deposition of various emissions caused by the combustion of fuels, abrasion of vehicle exploitation materials (mainly tires), road de-icing, and industrial processes [4]. GIS-based and multivariate statistical analyses have proved that elevated contents of Cd, Cu, Pb, and Zn in urban soils are commonly revealed from vehicle traffic, paint use, and other industrial discharges [5–7], whereas As and Hg primarily come coal burning coal [7,8]. Furthermore, the significant dispersion and deposition of metal-containing particulate matter could transport and redistribute PTEs in urban areas. This process is influenced by multiple physical and environmental factors, such as the urban landscape, wind direction, and urban water runoff [3]. More than 40 years ago, Sakagami et al. reported that there was a close relationship between heavy metal concentrations in soils and those in dust [8].

Many studies have shown that soil contamination with PTEs such as arsenic (As), cadmium (Cd), chromium (Cr), cobalt (Co), copper (Cu), lead (Pb), manganese (Mn), molybdenum (Mo), nickel (Ni), uranium (U), and zinc (Zn), has led to growing concerns regarding the severe negative effects on living organisms, including humans [9–11]. Undoubtedly, the human health in towns and cities is strongly dependent on the status of urban soils [12]. Pollution, compaction, loss of organic matter, changes in soil reaction, structural degradation, and infection by pathogenic microorganisms are only a few of the many adverse processes that affect and modify the ecological functions of urban soils [13–16]. On the other hand, PTEs in soils can generate airborne particles and dusts, which may affect the air's environmental quality [17–20]. In this way, the urban soils, especially in parks and residential areas that are not used for food crops, may also have a direct impact on public health, since PTEs can be easily transferred into human bodies [21–23]. In particular, the ingestion of dust and soil has been widely regarded as one of the key pathways by which children are exposed to the heavy metals and metalloids from paint, leaded gasoline, vehicles, and local industry [24,25].

The remediation of polluted soils is essential, and researchers continues to develop novel, science-based remediation methods. The risk assessment approaches are similar worldwide and consist of a series of steps to be taken to identify and evaluate whether natural or human-made substances are responsible for polluting the soil, and the extent to which that pollution is posing a risk to the environment and to human health [26].

The rehabilitation and restoration of land-based ecosystems is a key strategy for recovering the services (goods and resources) ecosystems offer to humankind [27]. Although the bioremediation of highly polluted soils with anthropogenic PTEs has been the focus of increasing research in the last few decades due to their toxic impacts on the environment [28–32], the urban soils that contain PTEs in lower doses (close to but not exceeding the levels established to protect the health of humans and the environment) have received scant attention.

However, most restoration and rehabilitation projects are focused on artificial, man-made, and high-maintenance strategies, which are costly and usually not successful over a longer period of time, as they depend on external inputs of energy and money, in addition to human management and control [33–35]. Increasingly expensive physical remediation methods such as chemical inactivation or sequestration in landfills are being replaced by science-based biological methods such as enhanced microbial degradation or phytoremediation. Restoration and rehabilitation strategies that are based on natural processes and cycles are sustainable, as they use natural flows of matter and energy, take advantage of local solutions, and follow the seasonal and temporal changes in the ecosystems [36]. Therefore, the use of nature-based solutions (NBSs) to restore degraded soil functions and improve soil quality could be a sustainable and successful strategy to enhance the ecosystem services [33,37–39]. By working together with the forces of nature, well designed measures require less maintenance, are more cost-effective, and if constructed in a good way, may even be more effective over a long period because nature's forces can increase the efficiency of the structure [33].

In this study, we aim to (i) evaluate the bioremediation capacity of some grasses and their suitability for lawn planting in settlements (in residential and non-residential areas, along roads, etc.); (ii) propose technological solutions for their practical application in an urban environment. Emphasis is placed on the potential of some perennial grasses and their application for the bioremediation of polluted urban soils, namely perennial ryegrass (*Lolium perenne* L.), crested wheatgrass (*Agropyron cristatum* L.), tall fescue (*Festuca arundinacea* Schreb), as well as bird's foot trefoil (*Lotus corniculatus* L.). A case study from the city of Plovdiv (Bulgaria) is presented together with an effective technological solution for the establishment of urban lawns and roadside green buffer patches.

## 2. Materials and Methods

### 2.1. Study Area

The city of Plovdiv is located at 24°45′ east longitude and 42°09′ north latitude, at an altitude of 160 m a.s.l. It is the second largest city in Bulgaria after Sofia, and is also one of the most densely populated cities in the country, with almost 450,000 inhabitants per 102 km$^2$ [40]. Inside the city proper are six syenite hills, several industrial zones, a densely populated central area, some moderately populated areas around it, wide network of busy streets and train tracks, big parks, and other green yards [41].

The climate in the city of Plovdiv is temperate, with mild influence from the Mediterranean Sea and a large temperature spread between summers and winters. The average annual temperature is 12.3 °C, with a maximum in July (32.3 °C) and minimum in January (6.5 °C). The average relative humidity is 73%. It is highest in December (86%) and lowest in August (62%). The total precipitation is 540 mm—the wettest months of the year are May and June, with average precipitation of 66.2 mm, while the driest is August, with an average of 31 mm. Gentle winds with speeds of up to 1 m s$^{-1}$ represent 95% of all winds during the year. The prevailing wind direction is from the west, rarely from the east. Mists are common in the cooler months, especially along the banks of the Maritsa River. On average there are 33 days with mist during the year (based on observations made in the period 1980–2010) [42].

Both the specific topography and microclimate peculiarities have caused worsened air quality (elevated PM10 and PM2.5 levels) in the city of Plovdiv (lack of winds, canyon street effect, etc.). The main inputs of atmospheric pollutants are from traffic emissions and residential heating, while the industrial sector has a minor impact [43–51]. An elevated content of PTEs in urban soils close to the transport infrastructure has been reported [41,52–55].

## 2.2. Experimental Design

On the basis of a literature search and our previous work, three perennial and one leguminous grass species were adopted as perspective candidates for construction of an urban lawn as NBS for sustainable soil management, namely perennial ryegrass (*Lolium perenne* L.), variety IFC Harmoniya; crested wheatgrass (*Agropyron cristatum* L.), variety Svejina; tall fescue (*Festuca arundinacea* Schreb), variety Albena; and bird's foot trefoil (*Lotus corniculatus* L.), variety Leo [56].

The city of Plovdiv is divided into six administrative areas according to the General Development Plan [55], so six experimental plots (one per area) were selected and involved in the study with the permission of municipal authorities. All experimental plots were in permanent grass areas (part of the green urban infrastructure), situated along the main arteries of the road network in the given administrative area. Each plot (1 m width and 5 m length) consisted of five square subplots measuring 1 m × 1 m, following the road direction. Four grass species were planted as monocultures in the experimental subplots (one species per subplot per administrative area) and the fifth subplot was set as a mixed stand (1:1:1:1) (Figure 1).

Sowing was performed in spring 2019 following a modified agro-technology approach, as explained in the Results and Discussion (Section 3.1). Weed control during the experiment was achieved through manual extraction in order to avoid compromising the results, but some effective approaches for sustainable weed management in urban lawns could be applied [56].

As all experimental plots were located in permanent grass areas belonging to the city's green infrastructure, it was possible to make real time observations in a real urban environment and to assess the complex effects of anthropogenic activities. Periodical observations and measurements of plant status and development were performed during the vegetation period. Experimental plots were treated and maintained by the municipality in the same manner as the whole green area they were a part of (irrigation, mowing, etc.).

## 2.3. Sample Collection

Soil samples from each subplot were taken before sowing, aiming to assess the concentrations of PTEs and the ecological status of urban soils at the beginning of the experiment. After 6 months (autumn of 2019), approximately 20 g aboveground and underground plant biomass samples were separately taken from each of the sampled subplots. Soil samples were also collected from the rhizosphere at each sampling point where vegetation was previously sampled. All plant and soil samples were labelled, placed in paper bags, and transported to the laboratory. Plant samples were carefully cleaned by soil particles, rinsed with deionized water, and air-dried. Soil samples were manually cleaned by removal of organic residues, sieved, and air-dried to constant weight.

## 2.4. Soil Microbiology Analyses

The effect of NBS on urban soil quality was assessed not only via the phytoremediation effect but also by evaluating some of the soil microbiota's properties. It is well known that these communities play a key role in biogeochemical cycles and directly affect soil processes and vegetation cover [57,58].

Rhizosphere soil samples from each subplot were collected in sterile containers and transported in dark and refrigerated conditions to the Department of Biochemistry and Microbiology at Plovdiv University "Paisii Hilendarski". Simultaneously, some rhizosphere soil samples from the surrounding grass area were also collected in order to be used as a background reference so as to evaluate the impacts of the newly established experimental plant species and varieties.

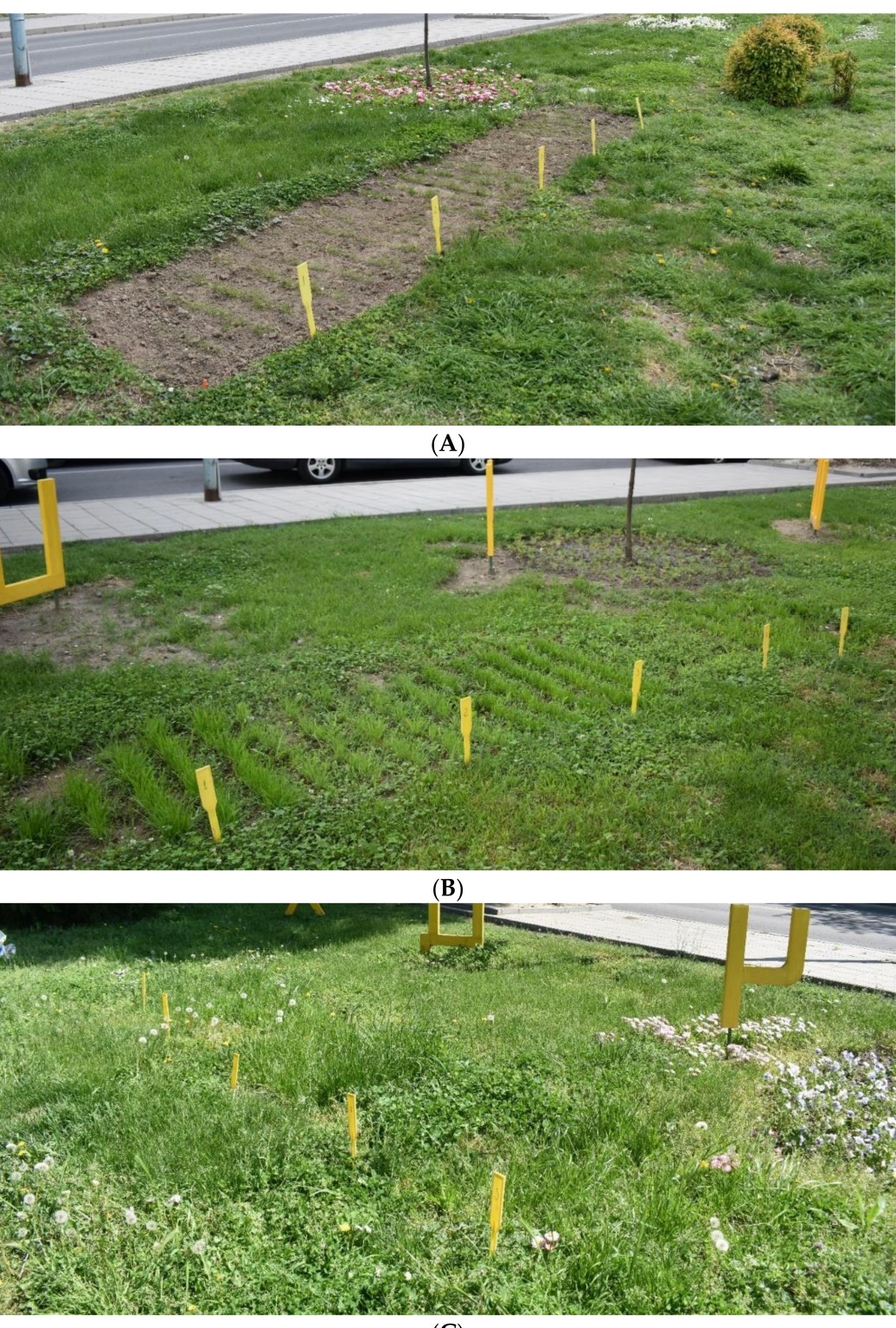

**Figure 1.** One of the experimental plots shown 30 days (**A**), 60 days (**B**), and 90 days (**C**) after sowing.

Microbiological analyses were scheduled to be performed within 24 h after sampling. For this purpose, 1 g of soil from each subplot was placed in 99 mL of sterile distilled water. The sample was homogenized for 30 min at 200 rpm to separate the cells from the soil particles. A semi-automatic Stuart SC6 + counter (BioCote, Coventry, UK) was used to count the colonies that formed. The number of viable microorganisms was determined by counting the number of colonies formed after inoculation on agar at 22 °C over a 7 day pe-

riod according to ISO 6222:1999. The total amount of soil fungi was determined by counting the colonies after superficial sowing with 100 μL pre-diluted and homogenized soil sample (×100) of Saburo selective medium with chloramphenicol and incubation for 14 days at 22 °C [59].

To assess the ability of microbial communities to absorb organic sources in their pure state, a physiological profile (CLPP) of microbial communities was prepared. The absorption patterns of individual carbon sources were determined using BIOLOG-EcoPlates (Biolog Inc., Hayward, CA, USA) containing 31 different C-sources in triplicate. The ability of microorganisms to absorb organic carbon was assessed by the rate of change in color of the indicator present in the plaques from colorless to purple. The inoculum was prepared by suspending 1 g of soil in 99 mL of sterile saline. The resulting suspension was homogenized for 30 min at 200 rpm on a rotary shaker, then allowed to settle for 10 min. This was filtered sequentially through 8.0 and 3.0 μm filters (Whatman, Maidstone, UK).

For inoculation and incubation, a paired culture with 100 μL suspension on R2A agar was performed to determine the number and inoculation of the plates with 120 μL filtrate in each well. The plates were incubated for 7 days at 22 °C, and the absorbance at 620 nm was measured every 24 h with a MULTISKAN FC microplate reader (Thermo Fisher Scientific, Shanghai, China) to determine growth kinetics.

The color of each plate was expressed as the average color of the wells (AWCD) according to the following formula:

$$\text{AWCD} = \sum \frac{(n_i - c)}{31} \tag{1}$$

where $n_i$ is the average absorption for the carbon source and $c$ is the control.

Data are presented as the mean of three replicates. $Ni\text{-}c \leq 0$ values were excluded from the analysis. The mean staining of each well in the plates (AWCD) was calculated daily to establish the kinetic profiles. Substrates were grouped into 6 categories, including carbohydrates, carboxylic acids, phenolic acids, amino acids, amines, and polyols, according to Kenarova et al. [60].

*2.5. Analytical Methods*

Concentrations of PTEs in both air-dried soil and plant samples were estimated after MW digestion in closed PTFE vessels (ETHOS ONE, Milestone, Shelton, WA, USA) by using a mixture of $HNO_3 + H_2O_2$. All reagents used were high-purity analytical grade. To control possible contamination due to the reagents or the sample preparation procedure, blank samples were prepared and analyzed for every sample batch. The calibration solutions for both ICP methods were prepared after appropriate dilution of Multi VI (30 elements in $HNO_3$, Merck, Darmstadt, Germany) traceable to NIST.

The total contents of Cu (324.754 nm), Mn (348.291 nm), Pb (220.353 nm), and Zn (206.200 nm) were determined via ICP-OES (iCAP 6300 Duo S, Thermo Scientific) at the appointed spectral lines by axial plasma observation, with the exception of Mn (for which the radial plasma view mode was used).

The total content of trace elements was determined via inductively coupled plasma mass spectrometry (ICP–MS) (Agilent 7700, Agilent Technologies, Inc., Santa Clara, CA, USA) by measuring the following isotopes: $^{75}$As, $^{111}$Cd, $^{59}$Co, $^{95,97}$Mo, $^{60,62}$Ni, $^{238}$U. For online correction of the non-spectral matrix effect in ICP–MS measurements, an internal standard ($^{103}$Rh) was applied.

Both analytical methods were validated (for the total content of elements) via analysis of certified reference materials (CRM). For soil analysis, a Loam Soil ERM—CC141 instrument was used for validation and quality control purposes. A portion of the CRM was digested and analyzed together with the samples. The measured concentrations of all tested elements were statistically comparable with the declared certified values (for extraction with Aqua Regia). Validation and quality control of the aboveground and underground plant samples were performed using the CRM—Bush Branches and Leaves NCS DC73348,

LGC standards. The percentages of recovery ranged from 91 to 99%, depending on each analyzed element. The extraction of trace elements and analyses were performed in the Laboratory of Analytical Chemistry and Computer Chemistry of the University of Plovdiv "Paisii Hilendarski".

### 2.6. BAF and TF

The bioaccumulation factor and translocation factor can be used to give clues on the suitability of a plant in phytoremediation. This is because phytoremediation technology uses the potential for heavy metal bioaccumulation and the exclusion of plants to cleanup areas polluted with heavy metal [52,61,62].

The bioaccumulation factor (BAF) of the studied PTEs was calculated using the ratio of the metal content in the roots to that in the soil using the formula used by Yoon et al. [63]:

$$\text{BAF} = \frac{PTE\ level\ in\ the\ root}{PTE\ level\ in\ the\ soil} \tag{2}$$

Bioaccumulation defines the ability of a plant to accumulate heavy metals, while allowing for their initial substrate content. The higher the values that it takes on, the higher the concentration of the element found in the plant biomass, as assessed against its initial substrate content. As the basis for assessing the bioaccumulation of metals, the four-degree scale was adopted [64]. According to this scale, a BAF of <0.01 means no accumulation, the range of 0.01–0.1 indicates low bioaccumulation, the range of 0.1–1.0 indicates medium bioaccumulation, and above 1.0 indicates high bioaccumulation.

Translocation factor (TF) was used as a measure of the element transport from the roots to the branches and was calculated through the formula used by Yoon et al. [63]:

$$\text{TF} = \frac{PTE\ level\ in\ the\ shoot}{PTE\ level\ in\ the\ root} \tag{3}$$

The translocation factor is a measure of the phytoextraction capacity of plants. It indicates that there is a possibility for the translocation of PTEs from the roots to the aboveground parts of plants. When TF > 1, it is obvious that the element is effectively transported from the roots to the aboveground plant organs.

### 2.7. Statistical Methods

A statistical evaluation was performed using the software Statistica 12.0 [65]. A descriptive analysis was performed using the arithmetic mean, median, range, standard deviation (SD), and coefficient of variation (CV). Relationships between the contents of individual elements in collected soil samples were tested using Pearson correlation coefficients ($p < 0.05$). The cluster analysis was carried out based on Ward's method, in which the similarity criterion is the squared Euclidean distance. The distance coefficients express the degree of similarity as distances in multidimensional space; thus, as the distance value decreases, the similarity increases.

## 3. Results and Discussion

### 3.1. Technological Solutions for the Establishment of Urban Lawns or for Recultivating Existing Urban Green Areas

3.1.1. Selection of Plant Species and Varieties

Perennial grasses (Poaceae family) are spread worldwide. They have high resistance to abiotic and biotic factors, ecological stability and adaptability, and are widespread in natural meadows and pastures, occupying up to 80–90% of grassland. They grow well both in the lowlands in rich soils and in the highlands in poor and unproductive areas.

Perennial and leguminous grasses are well known to improve the structure of the soil, protect it from erosion, and leave the most organic residues, which increases the humus content. The perennial grasses used in the present study are characterized by good productivity; high adaptability; and resistance to frequent mowing, grazing, and trampling.

Tall fescue (*Festuca arundinacea* Schreb) variety Albena is suitable for creating long-term lawns and green patches (9–10 years), especially for grassing areas subject to trampling. It withstands dry and wet soils, as well as saline and contamination with heavy metals. It can be used to create mixtures with alfalfa, sainfoin, and bird's foot trefoil. Monoculture is sown on swampy, saline, or contaminated soils. Moreover, it is a very suitable variety for areas at risk of soil erosion. *Festuca arundinacea* variety Rahela may absorb a relatively high amount of heavy metal ions from the soil, without significant reductions in biomass yield, which is expected in the case of bioremediation practices. Therefore, this cultivar could be regarded as a potential candidate for the phytoextraction of Cd-contaminated soil [66].

Zhang et al. [67] revealed that some perennial grasses accumulate higher amounts of lead (Pb) in their root tissues than in the aboveground biomass, with significant genotypic differences between different species. The results obtained in the experimental studies performed by many other authors were quite similar [68–70]. These authors also stated that in addition to the fact that perennial grasses have the ability to accumulate larger amounts of heavy metals in the roots or rhizomes, they could represent a vital part of the ecological system in the urban landscape and could have applications in the construction of parks and establishment of roadside green buffer patches and sports fields. Perennial grasses are strongly recommended due to the dense vegetation cover they form after mowing, their high anti-erosion potential, and the fact that their aboveground biomass is less flammable after drying when compared to other plants used for decoration and landscaping. The formed grass turf also improves the microclimate, helps to absorb carbon dioxide, increases biodiversity, and improves soil fertility in the area. Perennial ryegrass (*Lolium perenne* L.) variety IFC Harmoniya is suitable for creating urban lawns and green patches and for landscaping. It can be used as a component in mixtures with alfalfa, white clover, or bird's foot trefoil, and for decorative purposes in mixtures with red fescue.

Global warming is a serious reason to look for suitable species with increased adaptability and high productivity. Of interest are xeromesophytic and xerophytic grasses, which are tolerant to high summer temperatures and low soil moisture. These include the species of the genus *Agropyron*. In years with insufficient rainfall, they have an advantage over other perennial grasses and are extremely long-lasting. Crested wheatgrass (*Agropyron cristatum* L.) and standard wheatgrass (*Agropyron desertorum* Fisch.) are very suitable for the establishment of urban lawns. The variety Svejina is suitable for creating pastures and anti-erosion grasses and maintaining the landscape. It can be used as a component of different mixtures, especially in combination with white clover or bird's foot trefoil.

Leguminous grass species (Fabaceae family) are an economic source of soil nitrogen through the process of symbiotic nitrogen fixation. Thus, they help preserve and strengthen the soil's microbial environment and fertility.

Bird's foot trefoil (*Lotus corniculatus* L.) is a perennial legume, a natural tetraploid, and cross-pollinated species [71], which gives it the ability to adapt to soils with poor drainage in winter and limited water supply in summer. Due to its highly developed root system, bird's foot trefoil is not demanding of the soil or its fertility. This is confirmed by the natural habitats in which it occurs. The species is also known as an alternative crop for soils with a number of restrictions [72,73]. It tolerates lighter and poorer soils located in hilly areas, in the plains and mountains, as well as in areas with richer soils but with prolonged droughts. The growth of growing bird's foot trefoil in acidic soils is satisfactory. It is tolerant to overwatering, drought, and soil salinity; this gives it an advantage over other legume grasses, such as alfalfa (*Medicago sativa* L.) or white clover (*Trifolium repens* L.) [74–76]. It is drought-resistant, like alfalfa, and tolerates soil moisture, like red clover. It grows in peat soils and in periodically but not permanently flooded meadows. From this, it is clear that bird's foot trefoil can be successfully grown in fields and in mountainous and foothill conditions. Undoubtedly, the cultivation of legume species in grass mixtures provides advantages not only in protecting the soil from water and wind erosion and in the regulation of the water regime, but also in the N-fertilization of soil.

The advantages of bird's foot trefoil as an undemanding and responsive crop in different soil and climatic conditions make it one of the main legume components of grass mixtures used for grassing areas in urban environments, such as parks, inter-block spaces, and buffer areas around roads. The main advantages of grassing or renewal of the composition of grass species are the improvement in soil structure, prevention of compaction, and improvement in soil nutrient composition. Apart from maintaining the soil surface, as a component of grass mixtures it contributes to the mobilization of heavy metals such as Cu, Zn, and Mn and reduces concentrations in the horizon by 0–10 cm [77]. The different varieties of *Lotus corniculatus* show good development and opportunities for the accumulation of metals in the roots in the following order of decreasing concentration: Fe > Zn > Mn > Cu. *Lotus corniculatus* variety Leo was created in the USA and is a low-growing variety used for pastures and soil reclamation. This feature makes it a suitable plant species for the phytoremediation of soils and for reducing the toxicity of some heavy metals as a result of various pollutants from anthropogenic activities in the urban environment [78].

3.1.2. Technology for the Establishment of Urban Lawns and Green Buffer Patches

This technological model was validated in Bulgaria but could be easily adapted to other countries according to the local conditions. Bulgaria has five climate zones (and subclimatic regions), as well as over 20 soil types. This diversity is also the reason for the need to select suitable species for grassing, in accordance with the biological requirements.

In case of relatively weak weed infestation of the grass areas, shallow plowing (single) is carried out at a depth range of 6–8 cm, and then the main tillage of the soil is carried out at a depth range of 20–25 cm in order to plow the plant remains, after which harrowing is performed to break up the soil aggregates. The area is then cultivated into a "garden condition".

Pre-sowing of the soil is carried out in the spring at the earliest opportunity to level the area, to preserve soil moisture, to destroy weeds, and to create a dense bed at the same depth of sowing for the rapid and uniform germination of seeds. Two pre-sowing surface treatments is carried out (harrowing and cultivating or milling)—the first at a depth of 6–8 cm, and the second at the depth of the seed layer of 4–6 cm, either obliquely or perpendicular to the direction of sowing to align the area and bring the soil up to a "garden condition". Rolling before and after sowing is a mandatory step. This is performed with smooth or articulated rollers across the direction of the pre-treatment (Table 1).

**Table 1.** Calendar of the main agro-technical activities in creating grass stands (according to the climate in Bulgaria).

| Activities | Months from February (II) to October (X) | | | | | | | | |
|---|---|---|---|---|---|---|---|---|---|
| | II | III | IV | V | VI | VII | VIII | IX | X |
| Shallow plowing of the soil at a depth of 6–8 cm is mandatory and leads to the destruction of much of the weed vegetation and preservation of soil moisture | | | + | | | | ++ | | |
| Basic soil tillage (20 cm) | | | | | | + | | | ++ |
| Soil cultivation (4–8 cm) | ++ | ++ | | | | | + | | |
| Soil milling (4–8 cm) | ++ | ++ | | | | | + | + | |
| Rolling the soil before sowing | | ++ | | | | | | + | |
| Sowing | | ++ | | | | | | + | |
| Rolling the soil after sowing | | ++ | | | | | | + | |

Note: + spring sowing; ++ early autumn sowing.

When using the perennial grasses, fertilization is one of the important factors in the formation of dense turf grass. Therefore, it is necessary to study the soil reserves of main biogenic elements (N, P, K) in the forms that are available for plant uptake. Based on these data, the necessary fertilization approach and rates can be discussed. Nature-based approaches will always be preferred, including the use of green manure, mulch, and crop rotation. In most cases, none of these NBS are possible in urban areas, so conventional fertilizers are chosen as the only solution. Phosphorus and potassium fertilizers are applied once with the main tillage. Their quantity is determined depending on the stock in the soil and the duration of use of the grass. In phosphorus-poor and moderately potassium-rich soils, it is recommended to apply 280–300 kg/ha of the active ingredient $P_2O_5$ and 120–150 kg/ha of the active ingredient $K_2O$. Nitrogen fertilizers are imported depending on the type of development of perennial grasses, being either "winter" or "winter–spring". For "winter–spring" species, nitrogen fertilizers are imported as follows: 2/3 in the spring and 1/3 in the autumn. In species with a "winter" type of development, 2/3 of the nitrogen fertilizer is applied in the autumn in order to stimulate the intensity of tillering and the formation of more shortened vegetative shoots, on which the density of grass turf in the following year largely depends, and 1/3 is applied in the spring (Table 2). In case of unbalanced and intensive fertilization with nitrogen combined with precipitation, perennial grasses show a great tendency to freeze in areas with pronounced climatic anomalies.

**Table 2.** Fertilization rates and norms.

| Fertilization | Storage Fertilizer, Active Ingredients kg/ha | Fertilization during the Growing Season, Active Ingredients kg/ha | |
| --- | --- | --- | --- |
| | | Autumn | Spring |
| N | - | 50–70 | 50–80 |
| $P_2O_5$ | 280–300 | | |
| $K_2O$ | 120–150 | | |

Note: Fertilizer rates are indicative; their quantity is determined depending on the stock in the soil and the duration of use.

The most suitable period for sowing in non-irrigated conditions is the end of February and beginning of March, and in irrigated conditions with good rainfall is in late summer, namely August and September, depending on the climatic conditions of the country.

Spring sowing is done as soon as possible because delays lead to risks, as frequent spring droughts can compromise the germination and garnishing of grasslands. Sowing is performed in narrow rows (up to 12 cm) to form grass turf with a high percentage of soil surface coverage. The sowing depth can vary from 1.5 to 2.0 cm depending on the size of the seeds. Perennial cereal grasses are sown with different sowing rates depending on the type, germination rate, and quality of the seeds to ensure the optimal density of the grassland (Table 3). The sowing of perennial grasses is mechanized and carried out obliquely or perpendicular to the direction of the last tillage. All modern drills suitable for sowing small seeds can be used. Rolling of the area is mandatory, which helps ensure fast and uniform germination of the seeds and additional leveling of the terrain. Sowing in cold, wet, or dry soils leads to dilution, as some of the seeds die, which can severely compromise the formation of dense grass ground cover.

**Table 3.** Sowing rate.

| Plant Species and Varieties | Sowing Rate, kg/ha |
| --- | --- |
| Perennial ryegrass (*Lolium perenne* L.)—diploid (2n) | 100–150 |
| Perennial ryegrass (*Lolium perenne* L.)—tetraploid (4n) | 200–250 |
| Crested wheatgrass (*Agropyron cristatum* L.) Variety Svejina | 100–150 |
| Standard wheatgrass (*Agropyron desertorum* Fisch.) Variety Morava | 100–150 |
| Tall fescue (*Festuca arundinacea* Schreb) Variety Albena | 100–150 |
| Cocksfoot (*Dactylis glomerata* L.) Variety Dabrava | 100–150 |

### 3.2. Bioremediation Effect Assessment

Tables 4 and 5 present the levels of the studied PTEs in the rhizosphere soils, shoots, and roots for five experimental variants (4 monocultures and 1 mixed culture). The levels of accumulation of an individual element by the plants differed between species and also between subplots.

**Table 4.** Contents of Mn, Zn, Ni, Cu, and Pb in urban soils and plant roots and shoots (mg/kg).

| Species | Plot | Mn | | | Zn | | | Ni | | | Cu | | | Pb | | |
|---|---|---|---|---|---|---|---|---|---|---|---|---|---|---|---|---|
| | | soil | root | shoot | soil | root | shoot | soil | root | shoot | soil | root | shoot | soil | root | shoot |
| Perennial ryegrass (*L. perenne* L.)— Variety IFC Harmoniya | 1 | 720 | 434 | 116 | 83 | 191 | 67 | 39 | 18 | 5.5 | 27 | 19 | 9 | 26 | 12 | 4.3 |
| | 2 | 487 | 355 | 98 | 76 | 94 | 83 | 34 | 19 | 7.5 | 27 | 19 | 11 | 30 | 17 | 6.9 |
| | 3 | 632 | 531 | 94 | 116 | 171 | 72 | 17 | 10 | 2.6 | 34 | 26 | 9 | 35 | 23 | 4.9 |
| | 4 | 527 | 303 | 71 | 158 | 312 | 91 | 29 | 10 | 2.1 | 39 | 23 | 8 | 68 | 25 | 3.3 |
| | 5 | 572 | 436 | 130 | 97 | 206 | 49 | 37 | 17 | 6.1 | 25 | 20 | 9 | 28 | 14 | 4.1 |
| | 6 | 510 | 347 | 69 | 133 | 227 | 59 | 18 | 6 | 3.6 | 39 | 28 | 12 | 74 | 32 | 8.2 |
| | AV | 575 | 401 | 96 | 111 | 200 | 70 | 29 | 13 | 5.0 | 32 | 23 | 10 | 44 | 21 | 5.0 |
| Crested wheatgrass (*A. cristatum* L.) Variety Svejina | 1 | 813 | 753 | 190 | 92 | 134 | 75 | 44 | 31 | 6.6 | 30 | 28 | 12 | 30 | 26 | 6.3 |
| | 2 | 444 | 384 | 140 | 59 | 129 | 104 | 29 | 15 | 7.9 | 19 | 17 | 13 | 25 | 16 | 9.6 |
| | 3 | 677 | 398 | 130 | 125 | 191 | 90 | 18 | 5 | 3.7 | 44 | 24 | 14 | 44 | 20 | 9.4 |
| | 4 | 443 | 378 | 74 | 118 | 210 | 101 | 29 | 15 | 4.9 | 30 | 26 | 9 | 47 | 25 | 6.4 |
| | 5 | 654 | 598 | 69 | 90 | 110 | 80 | 39 | 31 | 1.9 | 26 | 21 | 9 | 31 | 25 | 3.2 |
| | 6 | 502 | 391 | 89 | 138 | 198 | 88 | 20 | 13 | 2.4 | 40 | 37 | 13 | 73 | 54 | 7.9 |
| | AV | 589 | 484 | 115 | 104 | 162 | 90 | 30 | 18 | 5.0 | 32 | 26 | 12 | 42 | 28 | 7.0 |
| Tall fescue (*F. arundinacea* Schreb) Variety Albena | 1 | 655 | 605 | 114 | 89 | 109 | 44 | 39 | 28 | 4.5 | 29 | 27 | 7 | 35 | 25 | 4.2 |
| | 2 | 442 | 301 | 110 | 55 | 287 | 39 | 29 | 14 | 6.3 | 18 | 23 | 9 | 22 | 12 | 7.0 |
| | 3 | 695 | 426 | 96 | 116 | 237 | 66 | 18 | 8 | 3.3 | 40 | 29 | 12 | 40 | 23 | 7.4 |
| | 4 | 500 | 359 | 80 | 114 | 235 | 72 | 29 | 16 | 3.2 | 29 | 26 | 9 | 42 | 25 | 4.3 |
| | 5 | 635 | 458 | 67 | 107 | 244 | 61 | 39 | 22 | 3.3 | 31 | 26 | 7 | 35 | 24 | 3.0 |
| | 6 | 555 | 418 | 121 | 177 | 237 | 74 | 21 | 10 | 5.5 | 55 | 42 | 17 | 101 | 66 | 20.0 |
| | AV | 580 | 428 | 98 | 110 | 225 | 59 | 29 | 16 | 4.0 | 34 | 29 | 10 | 46 | 29 | 8.0 |
| Bird's foot trefoil (*L. corniculatus* L.) Variety Leo | 1 | 501 | 380 | 108 | 66 | 72 | 43 | 19 | 13 | 5.6 | 22 | 23 | 11 | 22 | 13 | 6.1 |
| | 2 | 407 | 274 | 49 | 53 | 65 | 71 | 28 | 12 | 5.7 | 18 | 21 | 13 | 22 | 11 | 4.7 |
| | 3 | 689 | 403 | 168 | 115 | 84 | 73 | 18 | 7 | 6.2 | 41 | 26 | 18 | 45 | 21 | 12.0 |
| | 4 | 433 | 235 | 35 | 97 | 76 | 63 | 28 | 11 | 2.2 | 24 | 16 | 9 | 34 | 16 | 2.3 |
| | 5 | 631 | 440 | 73 | 172 | 94 | 52 | 39 | 23 | 6.3 | 34 | 24 | 11 | 37 | 22 | 3.8 |
| | 6 | 548 | 211 | 119 | 163 | 98 | 67 | 22 | 5 | 6.8 | 51 | 31 | 19 | 103 | 27 | 20.0 |
| | AV | 535 | 324 | 92 | 111 | 82 | 61 | 26 | 12 | 5.0 | 32 | 24 | 13 | 44 | 18 | 8.0 |
| Mixed stand | 1 | 510 | 373 | 72 | 89 | 199 | 50 | 20 | 13 | 3.1 | 25 | 20 | 7 | 26 | 14 | 3.0 |
| | 2 | 333 | 454 | 134 | 49 | 260 | 57 | 24 | 20 | 9.4 | 16 | 26 | 12 | 22 | 25 | 12 |
| | 3 | 621 | 548 | 129 | 116 | 168 | 68 | 16 | 12 | 4.7 | 37 | 36 | 14 | 39 | 30 | 10 |
| | 4 | 456 | 443 | 69 | 129 | 234 | 61 | 30 | 17 | 2.8 | 31 | 27 | 9 | 47 | 34 | 4.4 |
| | 5 | 679 | 586 | 96 | 96 | 198 | 53 | 42 | 27 | 5.3 | 30 | 26 | 8 | 33 | 23 | 4.1 |
| | 6 | 429 | 292 | 93 | 129 | 349 | 68 | 16 | 4 | 3.0 | 39 | 30 | 14 | 72 | 22 | 6.7 |
| | AV | 504 | 449 | 99 | 101 | 235 | 59 | 25 | 16 | 5.0 | 30 | 27 | 11 | 40 | 25 | 7.0 |

AV—Average of 6 subplots.

Generally, the Mn contents in studied urban soils varied between 333 mg/kg and 813 mg/kg, Zn varied between 49 mg/kg and 179 mg/kg, Ni varied from 14 mg/kg to 44 mg/kg, while Cu and Pb varied in the ranges of 16–55 mg/kg and 22–103 mg/kg, respectively (Table 4). The rest of the PTEs also demonstrated significant variability in the studied urban soils, as follows: Co ranged from 5.6 mg/kg to 11 mg/kg, Mo from 0.19 mg/kg to 0.92 mg/kg, As from 2.8 mg/kg to 5.9 mg/kg, Cd from 0.21 mg/kg to 0.77 mg/kg, and U from 1.7 mg/kg to 2.8 mg/kg (Table 5). The statistical evaluation revealed that strong dependencies and synergy existed between the studied PTEs. Positive correlations were found between the following elements: Mn–Co ($r = 0.754$, $p = 0.001$), Mn–As ($r = 0.744$, $p = 0.001$), Mn–U ($r = 0.444$, $p = 0.014$), Zn–Cu ($r = 0.878$, $p = 0.001$), Zn–Pb ($r = 0.813$, $p = 0.001$), Zn–Mo ($r = 0.806$, $p = 0.001$), Zn–As ($r = 0.504$, $p = 0.005$), Zn–Cd ($r = 0.839$, $p = 0.001$), Cu–Pb ($r = 0.839$, $p = 0.001$), Cu–As ($r = 0.643$, $p = 0.001$), Cu–Cd ($r = 0.744$, $p = 0.001$), Pb–Mo ($r = 0.953$, $p = 0.001$), Cd–Mo ($r = 0.801$, $p = 0.001$), Co–Ni ($r = 0.832$, $p = 0.001$), Co–As ($r = 0.331$, $p = 0.044$), Co–U ($r = 0.556$, $p = 0.001$). Cu and As

were the elements with the highest numbers of significant relationships found. Ni was the sole element forming negative correlations, with both PB (r = −0.394, *p* = 0.031) and Cu (r = −0.400, *p* = 0.028).

**Table 5.** Contents of Co, Mo, As, Cd, and U in urban soils and plant roots and shoots (mg/kg).

| Species | Plot | Co | | | Mo | | | As | | | Cd | | | U | | |
|---|---|---|---|---|---|---|---|---|---|---|---|---|---|---|---|---|
| | | soil | root | shoot | soil | root | shoot | soil | root | shoot | soil | root | shoot | soil | root | shoot |
| Perennial ryegrass (*L. perenne* L.)— Variety IFC Harmoniya | 1 | 10 | 4.3 | 1.1 | 0.39 | 3.5 | 2.6 | 5.5 | 2.3 | 0.41 | 0.25 | 0.27 | 0.11 | 2.6 | 1.9 | 0.66 |
| | 2 | 8.3 | 4.1 | 1.4 | 0.27 | 1.2 | 1.3 | 3.6 | 1.7 | 0.45 | 0.40 | 0.55 | 0.24 | 2.3 | 2.4 | 0.74 |
| | 3 | 7.7 | 5.3 | 1.0 | 0.37 | 0.9 | 1.3 | 4.9 | 3.5 | 0.43 | 0.39 | 0.29 | 0.06 | 2.1 | 1.9 | 0.33 |
| | 4 | 7.6 | 3.1 | 0.4 | 0.55 | 2.5 | 1.5 | 5.5 | 2.1 | 0.19 | 0.67 | 0.56 | 0.20 | 2.4 | 2.6 | 0.93 |
| | 5 | 8.9 | 4.8 | 1.3 | 0.29 | 1.5 | 1.1 | 4.1 | 2.0 | 0.36 | 0.36 | 0.26 | 0.07 | 2.0 | 1.6 | 0.31 |
| | 6 | 5.9 | 2.4 | 0.9 | 0.65 | 4.1 | 1.7 | 4.0 | 1.6 | 0.37 | 0.54 | 0.78 | 0.10 | 1.7 | 1.1 | 0.21 |
| | AV | 8.09 | 3.99 | 1.01 | 0.42 | 2.29 | 1.58 | 4.61 | 2.17 | 0.37 | 0.44 | 0.45 | 0.13 | 2.17 | 1.91 | 0.53 |
| Crested wheatgrass (*A. cristatum* L.) Variety Svejina | 1 | 11 | 7.6 | 1.5 | 0.44 | 0.7 | 1.1 | 5.9 | 4.4 | 0.61 | 0.32 | 0.26 | 0.10 | 2.5 | 2.2 | 0.87 |
| | 2 | 7.1 | 3.3 | 1.8 | 0.21 | 0.9 | 0.9 | 3.0 | 1.6 | 0.60 | 0.28 | 0.28 | 0.25 | 1.7 | 1.8 | 1.25 |
| | 3 | 7.8 | 3.2 | 1.6 | 0.49 | 1.0 | 0.9 | 5.2 | 2.2 | 0.74 | 0.46 | 0.25 | 0.11 | 2.5 | 1.3 | 0.49 |
| | 4 | 7.2 | 4.0 | 0.9 | 0.38 | 0.7 | 0.6 | 4.9 | 2.3 | 0.37 | 0.59 | 0.82 | 0.27 | 2.1 | 4.1 | 1.97 |
| | 5 | 9.7 | 8.1 | 0.5 | 0.27 | 0.7 | 0.7 | 4.3 | 3.7 | 0.18 | 0.35 | 0.28 | 0.14 | 2.2 | 2.4 | 0.29 |
| | 6 | 5.8 | 4.7 | 0.7 | 0.63 | 1.2 | 2.0 | 4.1 | 2.9 | 0.37 | 0.60 | 0.74 | 0.16 | 1.8 | 1.6 | 0.17 |
| | AV | 8.03 | 5.16 | 1.17 | 0.40 | 0.87 | 1.03 | 4.55 | 2.83 | 0.48 | 0.43 | 0.44 | 0.17 | 2.12 | 2.23 | 0.84 |
| Tall fescue (*F. arundinacea* Schreb) Variety Albena | 1 | 9.5 | 6.7 | 0.9 | 0.34 | 1.0 | 2.3 | 5.0 | 3.4 | 0.34 | 0.32 | 0.32 | 0.09 | 2.4 | 3.1 | 0.49 |
| | 2 | 7.3 | 2.8 | 1.3 | 0.19 | 4.0 | 3.9 | 3.3 | 1.2 | 0.46 | 0.27 | 0.39 | 0.16 | 1.9 | 1.9 | 1.05 |
| | 3 | 7.7 | 4.4 | 1.4 | 0.33 | 1.6 | 1.2 | 5.5 | 3.0 | 0.66 | 0.39 | 0.37 | 0.12 | 2.1 | 1.9 | 0.43 |
| | 4 | 7.3 | 4.1 | 0.6 | 0.42 | 1.5 | 1.5 | 3.9 | 2.1 | 0.24 | 0.56 | 0.53 | 0.19 | 2.8 | 2.8 | 0.69 |
| | 5 | 9.6 | 6.1 | 0.7 | 0.34 | 1.7 | 1.5 | 4.1 | 2.6 | 0.21 | 0.41 | 0.47 | 0.20 | 2.2 | 2.0 | 0.42 |
| | 6 | 6.6 | 3.8 | 1.8 | 0.79 | 1.6 | 1.6 | 5.0 | 2.8 | 0.81 | 0.77 | 0.69 | 0.20 | 2.1 | 1.4 | 0.42 |
| | AV | 8.00 | 4.65 | 1.13 | 0.40 | 1.90 | 2.02 | 4.48 | 2.52 | 0.45 | 0.45 | 0.46 | 0.16 | 2.24 | 2.18 | 0.58 |
| Bird's foot trefoil (*L. corniculatus* L.) Variety Leo | 1 | 6.3 | 3.2 | 1.5 | 0.26 | 5.9 | 9.4 | 3.7 | 1.9 | 0.65 | 0.21 | 0.28 | 0.09 | 1.8 | 3.0 | 0.85 |
| | 2 | 7.1 | 2.3 | 0.9 | 0.19 | 9.1 | 8.2 | 3.3 | 1.1 | 0.29 | 0.26 | 0.30 | 0.13 | 1.9 | 2.6 | 0.44 |
| | 3 | 7.9 | 3.9 | 2.6 | 0.34 | 3.9 | 6.7 | 5.5 | 2.8 | 1.21 | 0.35 | 0.25 | 0.12 | 2.0 | 1.6 | 0.61 |
| | 4 | 6.9 | 3.0 | 0.4 | 0.32 | 5.1 | 9.2 | 3.5 | 1.7 | 0.14 | 0.42 | 0.27 | 0.14 | 2.4 | 2.6 | 0.27 |
| | 5 | 9.7 | 6.1 | 1.2 | 0.37 | 5.0 | 13 | 3.9 | 2.6 | 0.33 | 0.42 | 0.38 | 0.11 | 2.2 | 2.2 | 0.32 |
| | 6 | 6.6 | 1.9 | 2.0 | 0.92 | 8.7 | 6.6 | 4.9 | 1.4 | 0.95 | 0.62 | 0.35 | 0.15 | 2.0 | 0.9 | 0.48 |
| | AV | 7.41 | 3.39 | 1.45 | 0.40 | 6.27 | 8.90 | 4.14 | 1.90 | 0.59 | 0.38 | 0.30 | 0.12 | 2.05 | 2.15 | 0.49 |
| Mixed stand | 1 | 6.5 | 3.4 | 0.7 | 0.30 | 1.2 | 1.7 | 3.8 | 1.9 | 0.29 | 0.27 | 0.31 | 0.11 | 2.2 | 2.0 | 0.36 |
| | 2 | 6.1 | 4.2 | 2.1 | 0.19 | 2.0 | 2.1 | 2.8 | 1.9 | 0.73 | 0.27 | 0.52 | 0.21 | 1.8 | 1.8 | 1.20 |
| | 3 | 7.3 | 5.8 | 1.9 | 0.33 | 1.1 | 1.1 | 5.2 | 4.1 | 0.88 | 0.39 | 0.40 | 0.13 | 2.1 | 2.2 | 0.54 |
| | 4 | 7.1 | 4.8 | 0.5 | 0.44 | 1.1 | 1.8 | 4.6 | 2.8 | 0.23 | 0.59 | 0.64 | 0.20 | 2.5 | 3.0 | 1.08 |
| | 5 | 10 | 7.2 | 1.2 | 0.40 | 1.8 | 2.3 | 4.4 | 3.0 | 0.36 | 0.37 | 0.48 | 0.17 | 2.3 | 2.1 | 0.39 |
| | 6 | 5.6 | 2.1 | 0.7 | 0.62 | 4.3 | 3.5 | 4.3 | 1.3 | 0.33 | 0.51 | 0.49 | 0.14 | 1.7 | 0.7 | 0.18 |
| | AV | 7.13 | 4.60 | 1.18 | 0.38 | 1.90 | 2.08 | 4.17 | 2.51 | 0.47 | 0.40 | 0.48 | 0.16 | 2.11 | 1.94 | 0.62 |

AV—Average of 6 subplots.

Plant uptake of PTEs from soil occurs either passively with the mass flow of water into the roots or through active transport crossing the plasma membrane of root epidermal cells. Under normal growing conditions, plants can potentially accumulate certain metal ions at an order of magnitude greater than the surrounding medium [79]. Based on the average values of PTEs in their underground and aboveground organs, the studied plant species could be arranged into the following descending orders:

Mn—Crested wheatgrass > Tall fescue > Perennial ryegrass > Bird's foot trefoil
Zn—Tall fescue > Perennial ryegrass > Crested wheatgrass > Bird's foot trefoil
Ni—Crested wheatgrass > Tall fescue > Perennial ryegrass > Bird's foot trefoil
Cu—Tall fescue > Crested wheatgrass > Bird's foot trefoil > Perennial ryegrass
Pb—Tall fescue > Crested wheatgrass > Perennial ryegrass > Bird's foot trefoil
Co—Crested wheatgrass > Tall fescue > Perennial ryegrass > Bird's foot trefoil
Mo—Bird's foot trefoil > Perennial ryegrass > Tall fescue > Crested wheatgrass
As—Crested wheatgrass > Tall fescue > Perennial ryegrass > Bird's foot trefoil
Cd—Perennial ryegrass = Crested wheatgrass = Tall fescue > Bird's foot trefoil
U—Crested wheatgrass > Tall fescue > Bird's foot trefoil > Perennial ryegrass.

Positive relationships in plant roots were found between the following elements: Mn–Co (r = 0.907, *p* = 0.001), Mn–Ni (r = 0.727, *p* = 0.001), Mn–As (r = 0.876, *p* = 0.001), Zn–Cd (r = 0.492, *p* = 0.006), Ni–Co (r = 0.841, *p* = 0.001), Ni–As (r = 0.495, *p* = 0.005), Ni–U (r = 0.437, *p* = 0.016), Cu–Pb (r = 0.848, *p* = 0.001), Cu–As (r = 0.394, *p* = 0.031), Cu–Cd (r = 0.498, *p* = 0.05), As–Pb (r = 0.366, *p* = 0.046), Pb–Cd (r = 0.633, *p* = 0.001), Co–As (r = 0.844, *p* = 0.001), U–Ni (r = 0.437, *p* = 0.016). These correlations were quite different from those obtained from the soil samples (*p* < 0.05). Negative dependencies were proved for Mo–Ni (r = −0.373, *p* = 0.043), Mo–Mn (r = −0.606, *p* = 0.001), Mo–Co (r = −0.555, *p* = 0.001), and Mo–As (r = −0.576, *p* = 0.001).

A plant's ability to accumulate metals from soils can be estimated using the BAF, while a plant's ability to translocate metals from the roots to the shoots can be measured using the TF. Enrichment occurs when a contaminant taken up by a plant is not degraded rapidly, resulting in accumulation in the plant. The process of phytoextraction generally requires the translocation of heavy metals to the easily harvestable plant parts, i.e., the shoots. By comparing BAF and TF, we can compare the ability of different plants to take up metals from soils and translocate them to the shoots. Tolerant plants tend to restrict soil–root and root–shoot transfers, meaning much less accumulation occurs in their biomass, while hyperaccumulators actively take up and translocate metals into their aboveground biomass. Plants exhibiting TF and particularly BAF values less than one are unsuitable for phytoextraction [80].

It should be noted that the accumulation factors for all examined PTEs were considerably higher for the roots than for aboveground parts, except for the molybdenum. The highest BAF (2.16–15.69) and TF (0.69–1.42) values were calculated for this element (Table 6).

**Table 6.** Bioaccumulation and translocation of PTEs in studied experimental variants (BAF and TF values above 1 are shown in bold font).

| Element | Perennial Ryegrass (*L. perenne* L.) Variety IFC Harmoniya | | Crested Wheatgrass (*A. cristatum* L.) Variety Svejina | | Tall Fescue (*F. arundinacea* Schreb) Variety Albena | | Bird's Foot Trefoil (*L. corniculatus* L.) Variety Leo | | Mixed Stand | |
|---|---|---|---|---|---|---|---|---|---|---|
| | BAF | TF | BAF | TF | BAF | TF | BAF | TF | BAF | TF |
| Mn | 0.70 | 0.24 | 0.82 | 0.24 | 0.74 | 0.23 | 0.61 | 0.28 | 0.89 | 0.22 |
| Zn | **1.81** | 0.35 | **1.56** | 0.55 | **2.05** | 0.26 | 0.74 | 0.75 | **2.32** | 0.25 |
| Ni | 0.46 | 0.34 | 0.62 | 0.25 | 0.56 | 0.27 | 0.46 | 0.47 | 0.63 | 0.3 |
| Cu | 0.71 | 0.43 | 0.81 | 0.45 | 0.86 | 0.36 | 0.75 | 0.56 | 0.92 | 0.39 |
| Pb | 0.47 | 0.26 | 0.66 | 0.26 | 0.63 | 0.26 | 0.42 | 0.44 | 0.62 | 0.27 |
| Co | 0.49 | 0.25 | 0.64 | 0.23 | 0.58 | 0.24 | 0.46 | 0.43 | 0.65 | 0.26 |
| Mo | **5.46** | 0.69 | **2.16** | **1.19** | **4.72** | **1.06** | **15.69** | **1.42** | **4.99** | **1.09** |
| As | 0.47 | 0.17 | 0.62 | 0.17 | 0.56 | 0.18 | 0.46 | 0.31 | 0.6 | 0.19 |
| Cd | **1.03** | 0.29 | **1.02** | 0.39 | **1.01** | 0.35 | 0.81 | 0.4 | **1.19** | 0.34 |
| U | 0.88 | 0.28 | **1.05** | 0.38 | 0.98 | 0.27 | **1.05** | 0.23 | 0.92 | 0.32 |

According to the scale used for assessment of the bioaccumulation capacity of the studied plant species, the results revealed that perennial ryegrass (*Lolium perenne* L.) variety IFC Harmoniya had strong bioremediation potential for Mo (5.46), Zn (1.81), and Cd (1.03), and medium bioaccumulation potential for the rest of the PTEs. Furthermore, the BAF values for U (0.88), Cu (0.71), and Mn (0.70) were quite promising in terms of its application in urban lawn establishment.

Crested wheatgrass (*Agropyron cristatum* L.) variety Svejina was found to be have high bioaccumulation potential for Mo (2.16), Zn (1.56), U (1.05), and Cd (1.02), and medium potential for the other studied PTEs, although the BAF values for Mn (0.82) and Cu (0.81) were quite elevated.

Tall fescue (*Festuca arundinacea* Schreb) variety Albena demonstrated strong bioaccumulation of Mo (4.72), Zn (2.05), and Cd (1.01), and promising potential for U (0.98), Cu (0.86), and Mn (0.74).

Bird's foot trefoil (*Lotus corniculatus* L.) variety Leo was a leader in terms of the accumulation of Mo (15.69) and U (1.05), but not so much as a bioaccumulator of the other PTEs. Promising BAF values were found for Cd (0.81), Cu (0.75), and Zn (0.74).

Data from the mixed cultivation of these four plant species showed that the simultaneous planting could enhance the bioaccumulation potential when compared to the monoculture. The highest BAF values were obtained for Zn (2.32), Cd (1.19), Mn (0.89), Cu (0.92), Co (0.65), and Ni (0.63) and were statistically significant for Zn, Cu, Co, and Cd ($p < 0.05$). This fact clearly demonstrated that the higher biodiversity of phytocenosis could not only increase the biodiversity of animals and microorganisms, but also has a positive influence on soil properties.

### 3.3. Effects of NBS on Soil Microbiota

Microorganisms are not just soil inhabitants, but play key roles in biogeochemical cycles and help maintain soil homeostasis [81]. The speed and efficiency of the processes depends on a number of characteristics of the microbial communities, such as the biomass, metabolic activity, and total number of viable heterotrophic microorganisms [82]. These parameters, in turn, are controlled by a number of environmental factors, such as the temperature and humidity, chemical composition, and physical properties of the soil [83,84]. Microorganisms respond quickly to changes in the environment due to their intense interaction with the environment [85], while changes in their biomass, metabolic activity, or community structure can be early signals of changes in the functioning of the entire ecosystem [82].

The total amount of heterotrophic bacteria determined by cultivation at 22 °C (TVC22) is an indicator of the degree of the anthropogenic load on the soil from easily digestible organic matter. The TVC22 values in the surface soil layer varied from $53 \times 10^4$ cfu·g$^{-1}$ in the background in autumn to $696 \times 10^4$ cfu·g$^{-1}$ in the analysis of newly constructed grassy areas along key boulevards in the city of Plovdiv (Figure 2). There were no significant differences between the two seasons, or between the background and experimental values.

Soil fungi are another essential element of the soil microbiome, as they are actively involved in the cycling of substances and energy in soil ecosystems [86]. Anthropogenically modified soil can transform the composition of its microbiota in terms of increasing or decreasing the share of one or another species of micromycetes, and the conditions of constant man-made soil contamination can cause the replacement of "safe" fungi with potentially pathogenic and epidemiologically unsafe ones for human and animal species.

The analysis of the microbiological data showed that the fungi form an essential component of the soil microbiota in the surface layers of soils. Their total counts varied from $8 \times 10^3$ to $85 \times 10^3$ cfu·g$^{-1}$ in the background grass areas in spring and autumn, respectively (Figure 3). In spring, higher abundance of fungi was observed in the newly planted areas compared to the background soils (parks) used as controls in the same season ($p = 0.00823$). Significant seasonal differences were also observed, with higher total fungi counts in the autumn samples for both the experimental plots ($p = 0.0246$) and the control background soils (0.0006).

Fungi play a major role in the decomposition and processing of organic matter (85% of soil organic matter is decomposed by the combined effects of fungi and bacteria), and the decrease in their quantity was one of the main reasons for the decrease in humus content [87] The higher abundance of fungi in the experimental plot soils compared to background soils (parks) is an indicator of the better condition of these plots, which confirmed the results reported by Sharkova [88], who stated that in urban soils subjected to a greater influence of anthropogenic pressure, there is a decrease in the total number and activity of microorganisms, and of the fungi/bacteria ratio in particular.

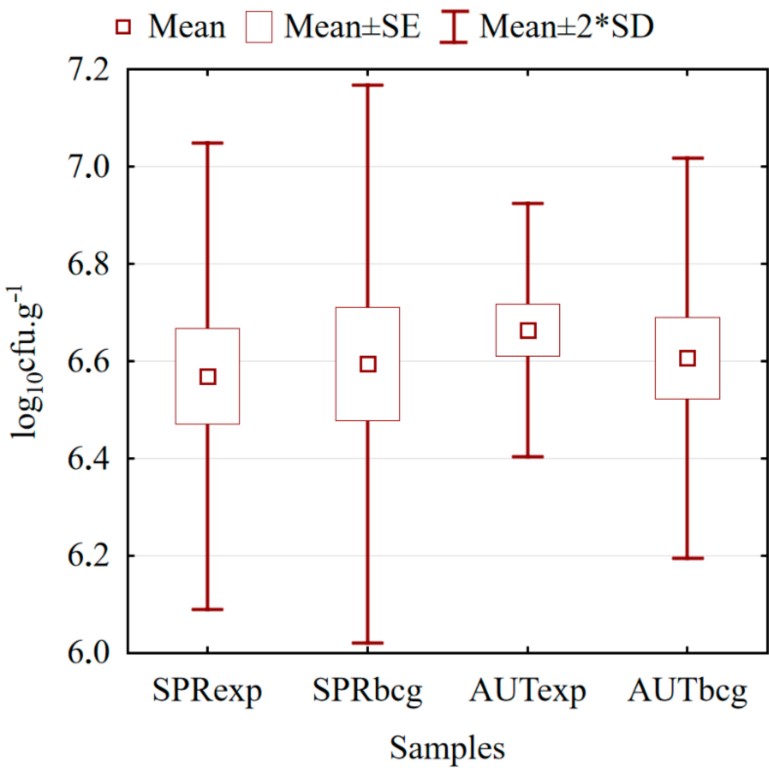

**Figure 2.** Average total heterotrophic bacterial count at 22 °C (TVC22) values in soil samples from experimental plots and background areas. Note: * SPRexp—experimental in spring; SPRbcg—background in spring; AUTexp—experimental in autumn; AUTbcg—background in autumn.

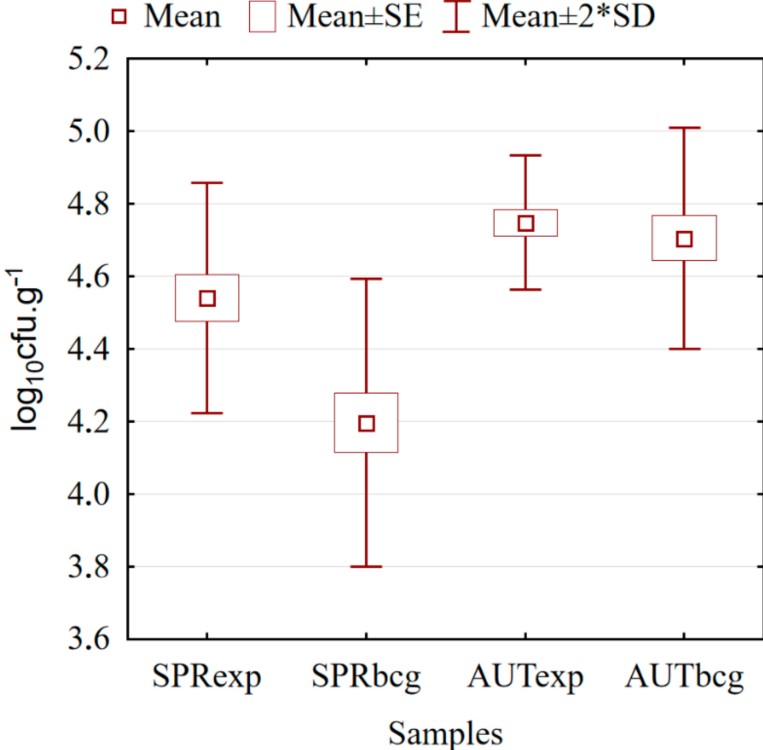

**Figure 3.** Average values of soil fungi in samples from experimental plots and background areas. Note: * SPRexp—experimental in spring; SPRbcg—background in spring; AUTexp—experimental in autumn; AUTbcg—background in autumn.

Anthropogenic stress and soil reclamation have direct impacts on the stability of microbial communities and their diversity [89], so in recent years physiological profiling at the level of microbial communities has been widely used to assess these indicators (CLPP). The analysis of the metabolic activity of soil microorganisms allows the study of the different characteristics of microbial communities. The EcoPlate test (BIOLOG) is designed specifically for research and ecological analysis of microbial communities. The ability to assimilate one or another substrate allows the qualification of microbial metabolic abilities, and accordingly the functional diversity of the microbial community. The results represent the physiological profile at the community level (CLPP) [90]. Stable and sustainable soils are defined as those with a high level of biological activity, high microbial diversity, and the ability to release nutrients from soil organic matter [57,91].

In the present study, the measurements were performed daily over a 7-day period. Based on the reported optical density (OD620) in the microtiter plates, the average staining value of each well (AWCD) was calculated for the samples in the initial and final study period. The AWCD growth curves over time for all samples were characterized by a short adaptation period (lag phase < 24 h). Differences in the nature of the curves were established; for the spring period they were of the sigmoidal type, while in autumn they were of the exponential type. The results did not show differences in the period of adaptation to the artificial environment (Figure 4A).

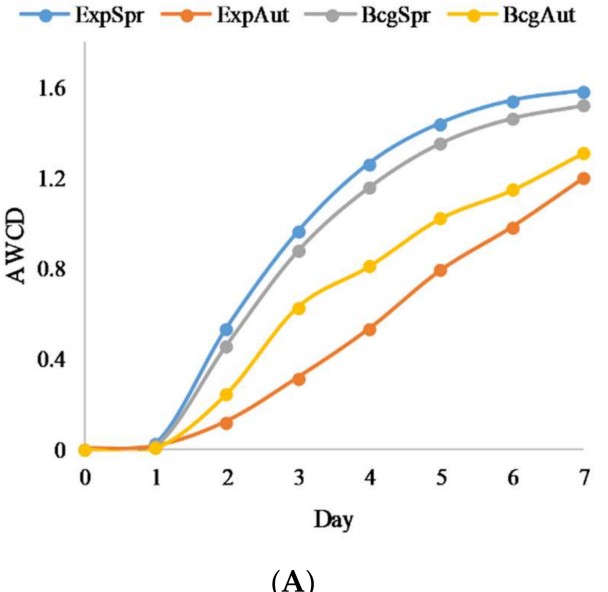

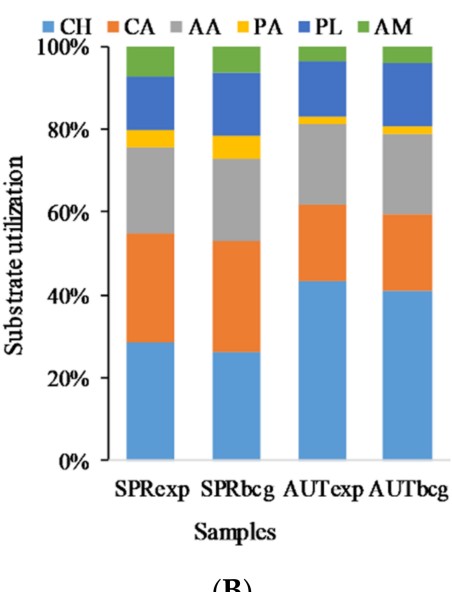

(**A**)     (**B**)

**Figure 4.** Kinetic curves of carbon uptake from communities in soil samples with different types of fertilization, expressed by AWCD (**A**), and bacterial activity, expressed by percentage uptake of substrates by category (**B**). Note: CH—carbohydrates; CA—carboxylic acids; AA—amino acids; PA—phenolic acids; PL—polyhydric alcohols; AM—amines.

According to Garland and Lehman [92], the slope of the inflection point curve between bacterial abundance and AWCD is a measure of community biodiversity. In this regard, the steeper the slope, the lower the biodiversity and vice versa [93]. The study revealed significant differences in the AWCD indicator values ($p < 0.01$) during the analyzed periods, as the values in the spring, reported in the background grass areas and in the experimental plots, were similar (Figure 4A). After the reclamation activities in the newly built green zones, the overall activity of the communities increased at the expense of their biodiversity. The value of AWCD 0.8 in the autumn period was reached within 96 h for the experimental variants and 120 h for the selected background zones ($p > 0.05$).

The results obtained were similar to the European database, according to which green areas have statistically lower average values compared to arable soils. At the same time, despite the large bioclimatic and soil differences, the differences between the categories were relatively small. One possible reason is the still small CLPP database [93].

The tests analyzed the levels of digestibility of 6 main groups of organic compounds, including polyhydric alcohols, carbohydrates, carboxylic acids, phenolic acids, amino acids, and amines, which are widely used as nutrients by soil bacteria, with the degree of their use varying depending on the total activity of the respective bacterial community. The levels of uptake by classes of substrates from the communities in the different variants are presented in Figure 4B.

Significant differences in the absorption levels of the studied groups of energy sources from different communities, as well as in the absorption of individual carbon substrates, were found. The number of absorbed substrates and the efficiency of absorption are defined by the environment and do not depend on the type of substrate. Again, for the studied periods no significant differences were found between the newly built green areas and their adjacent parks, or between the individual urban areas. In the samples from the spring period, the highest activity was found for carboxylic acids, followed by the activity levels for carbohydrates, amino acids, and polyhydric alcohols (polyols). Over the course of the experiment a shift in activity was found, which was associated with an increase in the absorption of carbohydrates and amino acids and a decrease in the absorption of carboxylic acids and polyols. The digestibility of amines and phenolic acids remained unchanged both between samples and between test periods.

During the autumn period, there was an increase in the number of assimilated substrates, especially in the samples from urban parks, as well as an increase in activity against easily digestible carbohydrates and carboxylic acids, which are the main products of carbohydrate metabolism. According to the results, microorganisms in urban soils are adapted to easily absorb available carbohydrates, but also have the potential to metabolize difficult-to-digest carbon sources.

The comparison of the physiological models of the communities through multivariate analyzes shows a clear differentiation between the initial and the final stage of the analysis, as well as between the background and experimental variants. The performed similarity analysis (ANOSIM), based on the study period, locations of analysis points, and types of green area, support the results. The obtained r-values are indicative of significant differences in the physiological profiles of the communities between the spring and autumn stages (r = 0.683, $p$ = 0.021), as well as between the samples from background and newly established experimental plots in autumn (r = 0.768, $p$ < 0.05). In the analyses performed at the beginning of the experiment, the communities from the background environment were characterized by higher levels of assimilation of the substrates, but no significant differences in the profiles were proven (r = 0.214, $p$ = 0.233). The higher (although not significant) activity in the spring in the background samples may have been a result of the presence of a better-developed root system. A number of authors have confirmed that root exudates in soils stimulate bacterial activity and diversity [94,95]. Catabolic activity and diversity values are indicative of high resilience to stressors and could recover more rapidly in the event of environmental disturbances [96].

The cluster analysis grouped CLPP profiles into three main clusters. The first (CA1) included samples mainly from the experimental variants during the autumn period, the second (CA2) grouped autumn samples from the background environment, and the third (CA3) combined spring samples (Figure 5). Overall, the results show that the establishment of new green areas (experimental plots) increases the physiological activity of bacterial communities and expands the range of digestible substrates.

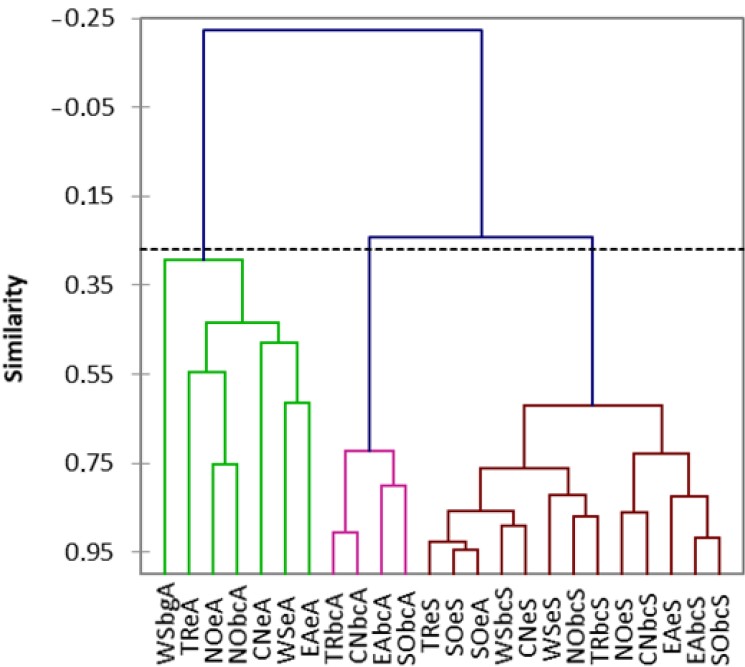

**Figure 5.** Hierarchical cluster analysis of substrates. Note: WS—administrative area 4 (west suburb); EA—administrative area 6 (east suburb); SO—administrative area 2 (south suburb); NO—administrative area 3 (north suburb); CN—administrative area 1 (central City Zone); TR—administrative area 5 (Trakiya suburb); bc—background; e—experimental; A—autumn; S—spring.

## 4. Conclusions

NBS solutions can be characterized by measures that enhance soil functions and soil resilience and are based on the concept of soil quality. Vegetation cover exerts various effects on soil structure (porosity, aggregate stability, organic matter content, water holding capacity) and soil microbial biocoenosis, that all create a more resilient soil ecosystem. More resilient systems have a buffer capacity against external impacts and create better livelihood conditions for above- and belowground life.

An effective technological solution for the establishment of urban lawns and roadside green buffer patches using perennial grasses was demonstrated. The establishment of grass-covered urban areas is a low-cost, ecologically friendly, and biodiversity-enhancing approach to the sustainable management of urban soils. All of these factors aim to enhance soil health and soil functions, through which local ecosystem services will be maintained or restored.

The present study could be used as an approach to modeling microbial communities by changing the purposes of the territories and establishing new green areas. The research was focused on the surface soil layer, as this area is most often subject to anthropogenic impacts. Biodiversity assessments were performed under similar conditions for all experimental variants, which is crucial in analyzing the effects of spatial variation and landscaping. There was a tendency to maintain high levels of biodiversity in the experimental areas when compared to the background areas, indicating that these soils can store more carbon.

As shown in the previous sections, NBS can have multiple environmental and social benefits, including via phytoremediation, supporting biodiversity, and enhancing soil processes and functions. In the context of ecosystem services, it could be stated that the sustainable management of urban soils is also crucial for carbon sequestration, water provision, and flood regulation, as well as for improving air quality and human health. All of these services are significantly enhanced when the NBS are integrated into urban planning and management.

The keys to working with NBS are the dynamics of the urban ecosystem and the system connectivity. Success lies in having a deep understanding of the processes and feedback systems that determine the environment–society relationships and the resilience of urban ecosystems. More resilient systems have a bigger buffer capacity against external disturbance and promote better livelihood for plants, animals, and people.

**Author Contributions:** Conceptualization, S.P., P.M.-S. and I.G.; methodology, P.P., V.S. (Veneta Stefanova), I.G., P.M.-S., D.G., V.S. (Violeta Stefanova) and I.I.; validation, V.S. (Violeta Stefanova), D.G., P.Z., E.V. (Ekaterina Valcheva), P.M.-S., I.G., E.V. (Evelina Varbanova), I.V., I.I., M.T., M.M., N.A., B.N. and S.P.; formal analysis, S.P., D.G., V.S. (Violeta Stefanova), E.V. (Evelina Varbanova), I.I., M.T. and M.M.; investigation, N.A., B.N. and S.P.; resources, I.I., P.M.-S. and I.G.; data curation, E.V. (Ekaterina Valcheva), G.H., N.A. and B.N.; writing—original draft preparation, P.M.-S. and S.P.; writing—review and editing, S.P.; visualization, S.P., I.I. and P.M.-S.; supervision, I.V. All authors have read and agreed to the published version of the manuscript.

**Funding:** This research was funded by Bulgarian National Science Fund, grant number KP 06 OPR 03/12.

**Institutional Review Board Statement:** Not applicable.

**Informed Consent Statement:** Not applicable.

**Data Availability Statement:** Not applicable.

**Conflicts of Interest:** The authors declare no conflict of interest.

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
