# Peer review of "Nature-Based Solutions for the Sustainable Management of Urban Soils and Quality of Life Improvements"

_land, doi:10.3390/land11040569_

Round 1

Reviewer 1 Report

Thank you for letting me review such an interesting and well-structured manuscript - congratulations, I find it suitable for publication in present form and of interest for the scientific community.

Author Response

Thank you very much!

Reviewer 2 Report

The manuscript in the present form does not address one of the declared research goals. 
The general organization and the writing are of enough clarity; minor edits could further improve its readability; authors can find in the attached report some general and specific remarks.
About the results, I would suggest carefully reviewing the organization of tables and the presentation of the outcome from statistical analyses as well as the presentation of findings.

Author Response

We have revised our manuscript according to the referees’ comments as follows:

Reviewer 2 recommendations

The manuscript’s aim is twofold in the research aspects – (i) assessing bioremediation capacity of some grasses and (ii) comparing it to other methods to show their superiority – and further offers a relevant proposal about the practical implementation of the considered solution in the context of the study.

Nonetheless, I have to admit that it appears as the second research goal has been only barely addressed, due to the intrinsic difficulty of consistently comparing “nature based solutions” to other technological approaches, and the lack of more concrete evidence reported in the manuscript. From this point of view, I would suggest reframing the in the abstract and the introduction and extending the discussion section about how an actual comparison could be performed – primarily this would require identifying the set of alternative technological solutions to be compared with the ones studied and proposed by the authors.

Authors answer: Thank you very much for all your recommendations! The second goal is deleted.

The general organization and the writing of the manuscript are of enough clarity; minor revision could further improve its readability; authors can find in the following some specific remarks (line numbers in the manuscript are on shown the right).

  1. Consider spread instead of range here: range between summers and winters. 127

Authors note: It is done

  1. Properly format –1 as an exponent: s–1 represent 95% of all 132

Authors note: It is done

  1. Specify the observation years on which the mentioned averages are based. year [42]. 135

Authors note: The period of observation was pointed out.

  1. which are the terms of the comparison (if any)?

revealed to a quite worsened air quality                                                                                               136-137

Authors note: The main atmospheric pollutants (PM10 and PM 2.5) were pointed out.

  1. Format properly i as an index where: ni - average absorption for carbon source; c - control.     225

Authors note: It is done

  1. Correct the misprint reagents

regents or sample preparation procedure, blank samples are pre-                                                          237

Authors note: It is done

  1. It sounds as a repetition, expunge the text:

Euclidean distance pro-vides a measure of the similarities between samples.                                293-294

Authors note: It is done

  1. I would include the subject: It

Withstands dry and wet soils, as well as saline or contaminated with heavy metals.                          312

Authors note: It is done

  1. I would add: Moreover, it is a

Very suitable variety for areas at risk of soil erosion.                                                                       314-315

Authors note: It is done

  1. I would insert in combination with

white clover or Bird's-foot trefoil.                                                                                                                      343

Authors note: It is done

  1. Put an appropriate reference (if the ellipsis was intended for such purpose) otherwise remove ‘(….)’. (....). 346

Authors note: It has been deleted.

  1. I would include the subject: It

Tolerates lighter and poorer soils located in                                                                                                   352

Authors note: It is done

  1. I would suggest step instead of “event” here:

event. It is performed with smooth or                    395

Authors note: It is done

  1. Please clarify what sole means (maybe Shallow plowing of the soil at a depth of 6-8 cm?) in Table 1. Calendar of the main agro-technical measures in creating grass stands (according to the climate in Bulgaria).         397-398

Activities

 The sole

Authors note: It is done

  1. I would suggest activities instead of “measures” here:

measures in creating grass stands (according to the                                                                                     397

Authors note: It is done

  1. Please, clarify the coding for the Table 1 – it would be beneficial to readability of the table also specifying the numbering used for months, e.g. Months from February (II) to October (X)

Authors note: It is done

  1. and correct the misprints: sowing

+ spring sawing

++ early autumn sawing

Authors note: It is done

  1. What activity is implied in the row: “Cultivation soil (4-8 cm)”?

Authors’ answer: We have in mind the soil cultivation process aiming to make it more suitable for planting. Cultivation improves soil structure by alleviating compaction, plus it offers the chance to apply fertilizers as it is explained below.

  1. I think a few remarks on the need for input should be addressed since the manuscript focus on “nature based solutions”: the use of fertilizers could imply chemicals and fossil energy.

When using the perennial grasses, fertilization is one of the important factors for the                      402

 formation of dense turf grass. Phosphorus and potassium fertilizers are applied                               403

Authors’ answer: Some explanations have been added.

  1. It would be better to avoid “variance” here, I could suggest variability / spread:

variance into the studied urban                                                                                                                         445

Authors note: It is done

  1. Being the correlation calculated for pairs of elements, it is not clear which are the significantly associated (linearly dependent) concentrations; consider the option of including a table or rephrase the statement to make it understandable:

positive correlations have been found between elements Mn-Co-As-U, Zn-Cu-Pb-Mo-As-Cd, as well as Co-Ni-Mn-As-U   (p<0.05)                                                                                                                            449-450

Authors answer: The statement was rephrased.

  1. I understand it should be included significant between “number of” and “relationships”:

Cu and As were the elements with highest number of relationships found. Ni                                     451

Authors note: It is done

  1. Check the number of significant digits and align properly the numbers in

Table 4. Content of Mn, Zn, Ni, Cu and Pb in urban soils and plant roots and shoots (mg/kg)                453

Authors note: It is done

  1. Again here, would authors clarify which are the actual pairs showing significant correlation values

 Positive relationships in plant roots have been found between elements Cu-Pb-As-                         475

Cd, As-Mn-Ni-Cu-Pb-Co, U-Ni, Zn-Cd and these correlations were quite different from                      476

those obtained by the soil samples (p<0.05). 477 25. Repetition of what already introduced above, consider deleting: A plant's ability to accumulate metals from soils can be estimated using the BAF, 479 which is defined as the ratio of metal concentration in the roots to that in soil. A plant's 480 ability to translocate metals from the roots to the shoots is measured using the TF, which 481 is defined as the ratio of metal concentration in the shoots to the roots. 482 26. Already included into the method section, consider deleting the parenthesis ><0.05)                                                                                                   477

Authors answer: The statement was rephrased.

  1. Repetition of what already introduced above, consider deleting:

A plant's ability to accumulate metals from soils can be estimated using the BAF,                              479

which is defined as the ratio of metal concentration in the roots to that in soil. A plant's                480

ability to translocate metals from the roots to the shoots is measured using the TF, which                481

is defined as the ratio of metal concentration in the shoots to the roots.                                              482

Authors note: It is done

  1. Already included into the method section, consider deleting the parenthesis

(p<0.05). 477 25. Repetition of what already introduced above, consider deleting: A plant's ability to accumulate metals from soils can be estimated using the BAF, 479 which is defined as the ratio of metal concentration in the roots to that in soil. A plant's 480 ability to translocate metals from the roots to the shoots is measured using the TF, which 481 is defined as the ratio of metal concentration in the shoots to the roots. 482 26. Already included into the method section, consider deleting the parenthesis (BAF >< 0.01 means no accumulation; 0.01–0.1—low bioaccumulation; 0.1– 496 1.0—medium bioaccumulation; and above 1.0—high bioaccumulation)

Authors note: It is done

  1. Here and in the following ‘significant’ means that a statistical test has been applied to compare the average BAF with the thresholds? Since it does not seem the case, I would suggest replacing the word; “a strong/ promising potential for” could suit.

significant 498

significant 506

significant 507

Authors note: It is done

  1. The following claim requires that background data (shown in Table 6) should be reported along with their standard deviation, and the number of replicates must be included when less that 6 (e.g., for missing sample or measurement) let alone that any implication of eventually excluding values of BAF and TF less than 1 should be properly discussed.

Data from the mixed cultivation of these four plant species showed that the simulta-                      512

neous planting could enhance the bioaccumulation potential when comparing to the mon-                513

oculture. Highest BAF values of Zn (2.32), Cd (1.19), Mn (0.89), Cu (0.92), Co (0.65) and Ni                514

(0.63) was obtained there being statistically significant for Zn, Cu, Co and Cd (p<0.05). 515 29. Please, use a consistent notation in the vertical axis label of Fig. 2 (the figure plots log 10 of TS22, as in the following Fig. 3). Please, also mind explaining if the intervals are calculated on the log scale or properly draw the two figures with estimates of mean, ±SE and ±2∙SD then apply the logarithm and draw it (nonsymmetrical intervals will result) and update the caption accordingly. Fig. 2. Average values of TVC22 in soil samples from experimental plots and back- 548 ground area 549 *SPRexp – experimental in spring; SPRbcg – background in spring; AUTexp – exper- 550 imental in autumn; AUTbcg – background in autumn. 551 30. Please, clarify what is significant (the difference between spring-autumn as well as backgroundgrass conditions?) Statistically significant values are found in experimental plots both 556 in the spring and in the autumn season (p><0.05). 557 31. Mind again that the graph in Fig. 3 does not allow performing visually such a comparison by means of the box (for confidence interval on the mean) and wiskers (for prediction interval on the concentration) Name properly the figure panes in Fig. 4 (on line 623) Fig. 4. Kinetic curves of carbon uptake from communities in soil samples with differ- 624 ent types of fertilization, expressed by AWCD (A) and bacterial activity, expressed by 625 percentage uptake of substrates by category (B). 626 32. I would suggest not significant instead of “unreliable” here: The higher (although 640 unreliable) activity in the spring in the background samples may be a result of the pres- 641 33. Maybe it would be clearer to specify that here the results are collectively commented: Overall, The results show that the establishment of new 649 green areas (experimental plots) increases the physiological activity of bacterial commu- 650 nities and expands the range of digestible substrates. 651 34. In the caption of Fig 5 on page 18 (lines 654-660) I would specify that the starred notes explain how the code for the units must be interpreted. Please, correct some misprint (it appears bc instead of bg many times) in the unit labels. Fig. 5. Hierarchical cluster analysis on the base of substrate *WS – Administrative area 4 (West suburb), EA – Administrative area 6 (East suburb), 655 SO – Administrative area 2 (South suburb), NO – Administrative area 3 (North suburb), 656 CN – Administrative area 1 (Central City Zone), TR – Administrative area 5 (Trakiya sub- 657 urb) 658 **bg – background; е – experimental 659 ***А – autumn; S – spring 35. The Conclusions sound too brief and somehow generic; the addition of a Discussion section would be beneficial to highlight the limits of the study as well as mentioning some expected future developments.><0.05)                      515

Authors’ answer: This statement is supported by the statistical evaluation of raw data obtained by all field experiments.

  1. Please, use a consistent notation in the vertical axis label of Fig. 2 (the figure plots log 10 of TS22, as in the following Fig. 3). Please, also mind explaining if the intervals are calculated on the log scale or properly draw the two figures with estimates of mean, ±SE and ±2∙SD then apply the logarithm and draw it (nonsymmetrical intervals will result) and update the caption accordingly.

 Fig. 2. Average values of TVC22 in soil samples from experimental plots and back-                           548

ground area                                                                                                                                                             549

*SPRexp – experimental in spring; SPRbcg – background in spring; AUTexp – exper-                          550

imental in autumn; AUTbcg – background in autumn.                                                                                 551

Authors note: It is done

  1. Please, clarify what is significant (the difference between spring-autumn as well as background grass conditions?)

Statistically significant values are found in experimental plots both                                                        556

in the spring and in the autumn season (p<0.05). 557 31. Mind again that the graph in Fig. 3 does not allow performing visually such a comparison by means of the box (for confidence interval on the mean) and wiskers (for prediction interval on the concentration) Name properly the figure panes in Fig. 4 (on line 623) Fig. 4. Kinetic curves of carbon uptake from communities in soil samples with differ- 624 ent types of fertilization, expressed by AWCD (A) and bacterial activity, expressed by 625 percentage uptake of substrates by category (B). 626 32. I would suggest not significant instead of “unreliable” here: The higher (although 640 unreliable) activity in the spring in the background samples may be a result of the pres- 641 33. Maybe it would be clearer to specify that here the results are collectively commented: Overall, The results show that the establishment of new 649 green areas (experimental plots) increases the physiological activity of bacterial commu- 650 nities and expands the range of digestible substrates. 651><0.05)                                                                                           557

Authors note: It is done

  1. Mind again that the graph in Fig. 3 does not allow performing visually such a comparison by means of the box (for confidence interval on the mean) and wiskers (for prediction interval on the concentration)

Name properly the figure panes in Fig. 4 (on line 623)

Fig. 4. Kinetic curves of carbon uptake from communities in soil samples with differ-                        624

ent types of fertilization, expressed by AWCD (A) and bacterial activity, expressed by                       625

percentage uptake of substrates by category (B).                                                                                         626

Authors note: It is done

  1. I would suggest not significant instead of “unreliable” here:

The higher (although                                                                                                                                             640

unreliable) activity in the spring in the background samples may be a result of the pres-                641

Authors note: It is done

  1. Maybe it would be clearer to specify that here the results are collectively commented: Overall,

The results show that the establishment of new                                                                                           649

green areas (experimental plots) increases the physiological activity of bacterial commu-                650

nities and expands the range of digestible substrates.                                                                                 651

Authors note: It is done

  1. In the caption of Fig 5 on page 18 (lines 654-660) I would specify that the starred notes explain how the code for the units must be interpreted. Please, correct some misprint (it appears bc instead of bg many times) in the unit labels.

Fig. 5. Hierarchical cluster analysis on the base of substrate

*WS – Administrative area 4 (West suburb), EA – Administrative area 6 (East suburb),                     655

SO – Administrative area 2 (South suburb), NO – Administrative area 3 (North suburb),                   656

CN – Administrative area 1 (Central City Zone), TR – Administrative area 5 (Trakiya sub-                  657

urb)                                                                                                                                                                            658

**bg – background; е – experimental                                                                                                               659

***А – autumn; S – spring

Authors note: It is done

  1. The Conclusions sound too brief and somehow generic; the addition of a Discussion section would be beneficial to highlight the limits of the study as well as mentioning some expected future developments.

Authors note: We have extended the Conclusion section.

Thank you for your consideration of this manuscript.

Sincerely,

Slaveya Petrova

Department of Ecology and Environmental Conservation, Faculty of Biology, Plovdiv University “Paisii Hilendarski”, Plovdiv, Bulgaria, e-mail:

Round 2

Reviewer 2 Report

The general organization and the writing of the manuscript have been improved in clarity; minor revision could further improve its readability; authors can find in the attached document reactions to their replies to my remarks (line numbers in the original manuscript are on shown the right).